# COMPOSITIONAL GAN (EXTENDED ABSTRACT): LEARNING IMAGE-CONDITIONAL BINARY COMPOSITION

**Samaneh Azadi, Deepak Pathak, Sayna Ebrahimi, Trevor Darrell**
University of California, Berkeley
`{sazadi, pathak, sayna, trevor}@eecs.berkeley.edu`

## ABSTRACT

Generative Adversarial Networks (GANs) can produce images of surprising complexity and realism but are generally structured to sample from a single latent source ignoring the explicit spatial interaction between multiple entities that could be present in a scene. Capturing such complex interactions between different objects in the world, including their relative scaling, spatial layout, occlusion, or viewpoint transformation is a challenging problem. In this work, we compose a pair of objects in a conditional GAN framework using a novel self-consistent composition-by-decomposition network. Given object images from two distinct distributions, our model can generate a realistic composite image from their joint distribution following the texture and shape of the input objects. Our results reveal that the learned model captures potential interactions between the two object domains, and can output their realistic composed scenes at test time.

## 1 INTRODUCTION

Conditional Generative Adversarial Networks have emerged as a powerful method for generating images conditioned on a given input, with the goal of learning a mapping from the source distribution to the output distribution. This involves transforming either a single object of interest (horses to zebras, label to image, etc.) or changing the style and texture of the input image (day to night, etc.). However, these direct transformations do not capture the fact that a natural image is a 2D projection of a *composition* of multiple objects interacting in a 3D visual world.

In this work, we explore the role of compositionality in the GAN framework and propose a new method which learns to map images of different objects sampled from their marginal distributions (e.g., chair and table) to a composite sample (table-chair) that captures their joint distribution of object pairs. For instance, given an image of a chair and an image of a table, our formulation is able to generate an image containing the same chair-table pair arranged naturally. To the best of our knowledge, our work is the first to address the problem of generating a composed image from two given inputs using a GAN, and can be trained under two paired and unpaired scenarios. In an unpaired training setup, one does not have access to the paired examples of same object instances with their combined compositional image. For instance, to generate a joint image from an image of a given table and a chair, we might not have any example of that particular chair beside that particular table while we might have images of other chairs and other tables together.

Our key insight is to learn to compose two objects by getting supervision from decomposing generated joint images back to the individual objects during training. This reformulation enforces a self-consistency constraint (Zhu et al., 2017) through a composition-by-decomposition (CoDe) network. We use this self-consistent CoDe network for an example-specific meta-refinement (ESMR) approach at test time to generate sharper and more accurate composite images: We fine-tune the weights of the CoDe network on each given test example by the self-supervision provided from the decomposition network.

## 2 COMPOSITIONAL GAN

Conditional GANs have been applied to several image translation problems such as day to night etc. (Isola et al., 2017). However, the composition problem is more challenging than just translating

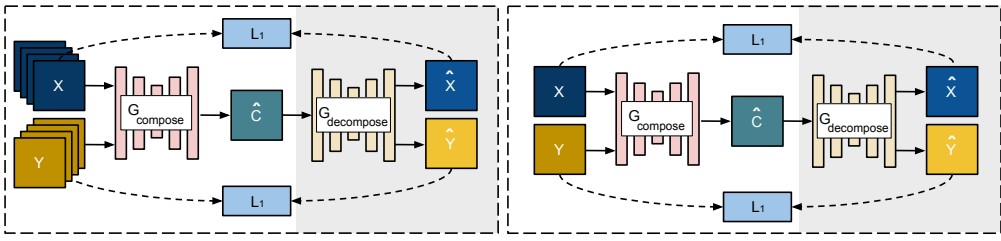

(a) Training Time: Decomposition As a Supervisory Signal   (b) Test Time: Decomposition for Example-Specific Meta Refinement

Figure 1: (a) CoDe Network is trained on all training images, (b) Test time ESMR step only uses one test example of $X$ and one test example of $Y$. The composition and decomposition generators are presented in pink and yellow, respectively.

images from one domain to another because the model additionally needs to handle the relative scaling, spatial layout, occlusion, and viewpoint transformation of the individual objects to generate a composite image. We now discuss our proposed Compositional GAN framework for generating a composite image given two individual object images. Let $x$ be an image containing the first object category, $y$ be an image of the second object, and $c$ be the image from their joint distribution. During training, we are given datasets $X = \{x_1, \cdots, x_n\}$ and $Y = \{y_1, \cdots, y_n\}$ from the marginal distribution of the two objects, as well as $C = \{c_1, \cdots, c_n\}$ from their joint distribution with images containing both objects. We further assume that the segmentation masks of objects are available for both individual images in $X, Y$ as well as the composite images in $C$. Our proposed binary compositional GAN model is conditioned on *two* input images $(x, y)$ in order to generate an image from the target distribution $p_{\text{data}}(c)$. The goal is to ensure that the generated composite image $\hat{c}$ contains the objects in images $x, y$ with the same color, texture, and structure while also looking realistic with respect to set $C$. Note that instead of learning a generative model of all possible compositions, our aim is to learn a mode of the distribution.

## 2.1 Self-Consistent Composition-by-Decomposition (CoDe)

The central idea of our approach is to supervise the composition of two images $x$ and $y$ via a self-consistency loss function that ensures that the generated composite image, $\hat{c}$, can further be decomposed back into the respective individual object images. The composition is performed using a conditional GAN, $(G_c, D_c)$, that takes the two RGB images $(x, y)$ concatenated channel-wise as the input to generate the corresponding composite output, $\hat{c}$, with the two input images appropriately composed. This generated image will be then fed into a another conditional GAN, $(G_{\text{dec}}, D_{\text{dec}})$, to be decomposed back into its constituent objects, $\hat{x}$ and $\hat{y}$ using a self-consistency $L_1$ loss function. The schematic of our self-consistent CoDe network is illustrated in Figure 1-(a).

In addition to the decomposition network, the generated composite image is also given to a mask prediction network, $G_{\text{dec}}^M$, that predicts the probability of each pixel in the composite image to belong to each of the input objects or background. A GAN loss with a gradient penalty (Gulrajani et al., 2017) is applied on top of the generated images $\hat{c}, \hat{x}^T, \hat{y}^T$ to make them look realistic in addition to multiple $L_1$ loss functions penalizing the deviation of generated images from their ground-truth. Furthermore, in practice, we explicitly model the scale and shift of objects by including a spatial transformer network (STN) (Jaderberg et al., 2015) before the composition layers.

In summary, the objective function for the full end-to-end model is:

$$
\begin{aligned}
\mathcal{L}(G) \quad = \quad & \lambda_1[\mathcal{L}_{L_1}(G_c) + \mathcal{L}_{L_1}(G_{\text{dec}}) + \mathcal{L}_{L_1}(\text{STN})] + \lambda_2 \mathcal{L}_{\text{CE}}(G_{\text{dec}}^M) \\
+ \quad & \lambda_3[\mathcal{L}_{\text{cGAN}}(G_c, D_c) + \mathcal{L}_{\text{cGAN}}(G_{\text{dec}}, D_{\text{dec}})],
\end{aligned}
$$

Here, $\mathcal{L}_{\text{CE}}(G_{\text{dec}}^M)$ is a cross-entropy loss for mask prediction. For more details of our model, one can refer to (Azadi et al.)[1].

We train Compositional GAN in two scenarios : (1) when inputs-output are *paired* in the training set, i.e., each composite image in $C$ has corresponding individual object images in $X, Y$, and (2) when training data is *unpaired*, i.e., images in $C$ do not correspond with images in $X$ and $Y$. Given the segmentation masks of the real composite images, we convert the unpaired data to a paired one

---

[1]citation allowed per communication with the workshop chairs.

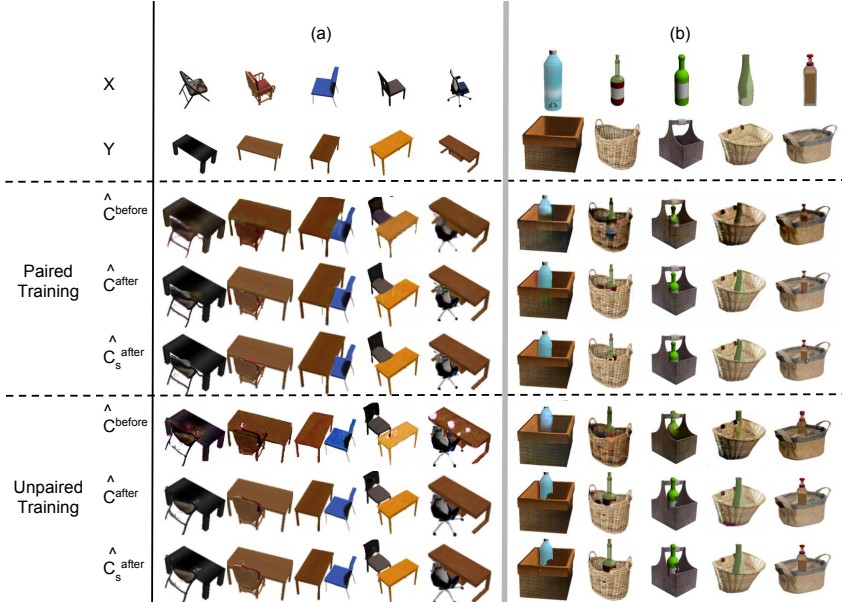

Figure 2: Test results on the chair-table (a) and basket-bottle (b) composition tasks trained with either paired or unpaired data. Details are provided in 2.2.

by cutting out the respective object segments from each composite image in $C$ to get the corresponding paired individual object images. Although these new object cutouts would be paired with the composite image, they are incomplete and not amodal because of occlusion in the composite image. Hence, we synthesize the missing part of these individual object cutouts using self-supervised inpainting networks (Pathak et al., 2016) which are trained on object images from $X$ and $Y$.

## 2.2 EXAMPLE-SPECIFIC META-REFINEMENT (ESMR)

The compositional GAN model not only should learn to compose two object with each other, but it also needs to preserve the color, texture and other properties of the individual objects in the composite image. While our framework is able to handle the former, it suffers at times to preserve color and texture of held-out objects at test time. We propose to handle this issue by performing per-example refinement at test time. Since our training algorithm gets supervision by decomposing the composite image back into individual objects, we can use the same supervisory signal to refine the generated composite image $\hat{c}$ for unseen test examples as well. Hence, we continue to optimize the network parameters using the decomposition of the generated image back into the two test objects to remove artifacts and generate sharper results. This example-specific meta-refinement (ESMR), depicted in Figure 1-(b), improves the quality of the composite image at inference.

Given the real samples from our training set, a GAN loss is also applied on the generated output resulting in the following loss function for refining the network:

$$\mathcal{L}(G) = \lambda(\|\hat{x}^T - x^T\|_1 + \|\hat{y}^T - y^T\|_1 + \|\hat{M}_x \odot \hat{c} - \hat{M}_x \odot x^T\|_1 + \|\hat{M}_y \odot \hat{c} - \hat{M}_y \odot y^T\|_1)$$
$$+ \quad [\mathcal{L}_{\text{cGAN}}(G_c, D_c) + \mathcal{L}_{\text{cGAN}}(G_{\text{dec}}, D_{\text{dec}})],$$

where $\hat{x}^T, \hat{y}^T$ are the generated decomposed images, $x^T$ and $y^T$ are the transposed inputs by the STN, and $\hat{M}_x, \hat{M}_y$ are the predicted segmentation masks. The decomposition and mask prediction networks reinforce each other in generating sharper outputs and predicting more accurate segmentation masks.

In the experiments of Section 3, we will present: (1) images generated directly from the composition network before and after this ESMR step, represented as $\hat{c}^{\text{before}}$ and $\hat{c}^{\text{after}}$, repectively, (2) images generated directly based on the predicted segmentation masks as $\hat{c}_s^{\text{after}} = \hat{M}_x \odot x^T + \hat{M}_y \odot y^T$.

## 3 EXPERIMENTS

In this section, we study the performance of our model with either paired or unpaired training data through multiple qualitative and user-study experiments on both synthetic and real datasets. First,

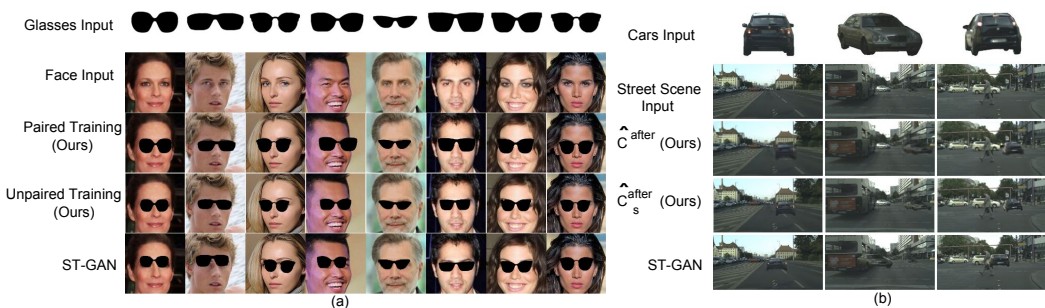

Figure 3: (a) Test examples for the face-sunglasses (a) and street scene-car (b) composition tasks.

we use the Shapenet dataset to study two composition tasks: (1) a chair next to a table, (2) a bottle in a basket. On the synthetic chair+table experiment, we deal with all four composition challenges, i.e., spatial layout, relative scaling, occlusion, and viewpoint transformation. In the basket+bottle experiment, the main issue is predicting the correct occluding pixels as well as the relative scaling of the two objects. The results are illustrated in Figure 2.

Second, we show our model performing equally well when one object is fixed and the other one is relatively scaled and linearly transformed to generate a composed image. In Figure 3, we present our results on the CelebA dataset composed with sunglasses downloaded from the web, as well as the Cityscapes dataset to compose a given street scene with a car image. Here, the problem is similar to the case studies of ST-GAN (Lin et al., 2018) with a background and foreground object.

In addition, we compare the performance of our model with ST-GAN based on a Mechanical Turk Study. In the unpaired face+sunglasses and the street scene+car composition tasks, 73% and 61% of the users, respectively, preferred our compositions to ST-GAN outputs. This confirms the superiority of our network to the state-of-the-art model in composing a background image with a foreground. More experimental results can be found in (Azadi et al.).

## 4 CONCLUSION AND FUTURE WORK

We proposed a novel Compositional GAN model addressing the problem of object composition in conditional image generation. We use a decomposition network as a supervisory signal to improve the task of composition both at training and test times. We evaluated our compositional GAN through qualitative experiments and user evaluations for both paired and unpaired training scenarios. In the future, we plan to extend this work toward modeling photometric effects (e.g., lighting) in addition to generating images composed of multiple (more than two) and non-rigid objects.

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
