# OpenReview forum: "Compositional GAN (Extended Abstract): Learning Image-Conditional Binary Composition"
_ICLR.cc/2019/Workshop/DeepGenStruct — DeepGenStruct 2019_

### Official Review · AnonReviewer2 · 2019-04-11
**clear and interesting submission**

**Rating:** 5
**Confidence:** 2

**Review:**

The authors propose a loss function to generate natural images that include two separate objects via a GAN.  The loss uses a sort of self-supervision by noting that the decomposition of a natural image with two objects into individual object images should match closely to the original object images.  The decomposition and composition network are then learned jointly.  Additionally, at test time, the authors provide a loss that tunes pixels to preserve color and texture.

I thought this short paper was quite clear (given space constraints) --- the objective was presented and described well.

The authors claim originality that the composition self-consistency loss is a new insight --- I am not familiar with work that conflicts with that claim, though I cannot be certain.

Questions/comments
- What are the qualitative and quantitative differences between the $\hat{c}^{after}$ and $\hat{c}^{after}_s$ images?  This should be made a bit more clear in the text.
- In the CelebA + Glasses experiment, what were the composite images used to train?

---

### Official Review · AnonReviewer1 · 2019-04-17
**interesting problem and reasonable approach**

**Rating:** 4
**Confidence:** 2

**Review:**

The paper tackles the problem of combining two images into one in a sensible way. In particular, the inputs are two objects (e.g., a bottle and a basket) and the output is an image containing both objects (e.g., a bottle in the basket). The challenge is that there aren't enough paired inputs and output. The authors proposed to 1) generate noisy examples by segmentation and inpainting and 2) adding a "self-consistency" loss to encourage that objects occurred in the inputs also occur in the output and segments in the output is close to the corresponding objects in the input. The "self-consistency" loss is also applied at test time to refine the output.

I find the problem pretty interesting. The model needs to learn the relative positions of the two objects as well as proper occlusion. The approach is pretty reasonable as well. I only have a couple questions / comments below.

- Aside from the paired examples, there is nothing in the loss function encouraging the model to *compose* the inputs with natural occlusion and position etc. So I'd like to see how many paired examples are needed to achieve the reported results, and how the results change with varying numbers of paired examples.

- It would be cool to control the relative position of the two objects.

---

### Decision · Program_Chairs · 2019-04-19
**Acceptance Decision**

Accept